# Waste Coffee Ground Biochar: A Material for Humidity Sensors

**DOI:** 10.3390/s19040801

**Published:** 2019-02-15

**Authors:** Pravin Jagdale, Daniele Ziegler, Massimo Rovere, Jean Marc Tulliani, Alberto Tagliaferro

**Affiliations:** 1Center for Sustainable Future Technologies, Italian Institute of Technology (IIT), Via Livorno 60, 10144 Torino, Italy; 2Department of Applied Science and Technology (DISAT), Politecnico di Torino, Corso Duca degli Abruzzi, 24, 10129 Torino, Italy; daniele.ziegler@polito.it (D.Z.); massimo.rovere@polito.it (M.R.); jeanmarc.tulliani@polito.it (J.M.T.); alberto.tagliaferro@polito.it (A.T.); 3INSTM R.U PoliTO-LINCE Laboratory, Department of Applied Science and Technology, Politecnico di Torino, Corso Duca degli Abruzzi, 24, 10129 Torino, Italy; 4Faculty of Science, University of Ontario Institute of Technology, Oshawa, ON L1H7K4, Canada

**Keywords:** waste coffee ground, biochar, humidity sensor, pyrolysis, impedance

## Abstract

Worldwide consumption of coffee exceeds 11 billion tons/year. Used coffee grounds end up as landfill. However, the unique structural properties of its porous surface make coffee grounds popular for the adsorption of gaseous molecules. In the present work, we demonstrate the use of coffee grounds as a potential and cheap source for biochar carbon. The produced coffee ground biochar (CGB) was investigated as a sensing material for developing humidity sensors. CGB was fully characterized by using laser granulometry, X-ray diffraction (XRD), Raman spectroscopy, field emission-scanning electron microscopy (FESEM), X-ray photoelectron spectroscopy (XPS), thermogravimetric analysis (TGA) and the Brunnauer Emmett Teller (BET) technique in order to acquire a complete understanding of its structural and surface properties and composition. Subsequently humidity sensors were screen printed using an ink-containing CGB with polyvinyl butyral (PVB) acting as a temporary binder and ethylene glycol monobutyral ether, Emflow, as an organic vehicle so that the proper rheological characteristics were achieved. Screen-printed films were the heated at 300 °C in air. Humidity tests were performed under a flow of 1.7 L/min in the relative humidity range 0–100% at room temperature. The initial impedance of the film was 25.2 ± 0.15 MΩ which changes to 12.3 MΩ under 98% humidity exposure. A sensor response was observed above 20% relative humidity (RH). Both the response and recovery times were reasonably fast (less than 2 min).

## 1. Introduction

Carbon might be the most widely-used material in sensor applications. It has the combined properties of a porous surface, electrical conductivity, biocompatibility, and chemical and electrochemical stability [1,2]. Many studies have shown that the electrical properties of carbon nano- or micro-sized materials as a sensor electrode have the ability to efficiently sense surface changes [3,4,5,6]. The integration of carbon nanomaterial into electrodes has been challenging due to cost and toxicity issues [7,8,9]. On the contrary biomass products like coffee waste is a qualified carbon raw material for the synthesis of valuable carbon materials because it is available in high quality and huge amounts, and it is an environmentally friendly renewable resource [10].

Coffee is one of the most important commodities and widely consumed beverages all around the world and grown in about 80 countries [11]. The global production has reached 11–13 million tons [12,13]. It is estimated that only in Italy more than 14 billion expressos are consumed every year and the number is growing continuously [14]. There are many ways to distinguish coffee such as origin, botanical variety, growing conditions, physiology, flavour, roasting, pre- and post-processing etc. [15]. The chemical composition (more than 1000 chemicals) [16] of coffee depends on physiological aspects such as degree of maturation [17,18,19]. Complex reactions take place during roasting at high temperatures and modify considerably coffee’s chemical composition, with some beneficial compounds are degraded and some others created. More than 950 compounds have been identified after roasting [20].

On the other hand, an estimated 380,000 tons of brewed coffee waste is discarded each year [16]. The pollution caused by coffee waste has also been a concern due to the high content of organic matter like caffeine, free phenols and tannins (polyphenols), and acid content, which are known to be very toxic to many life processes [21]. Coffee waste constitutes a source of a serious environmental problems in coffee-using countries [22]. 

There has been a significant amount of research activity going on in the area of waste coffee and its by-products utilisation during the last years. Numerous studies on the use of coffee waste lead to the conclusion that coffee by-products and wastes can be used in a variety of ways [23]. Most of the work is directed towards the use of this waste as starting material for the production of bioactive compounds, antioxidants, food additives, feeds, beverages, vinegar, biogas, caffeine, pectin, pectic enzymes, protein, and compost [24,25,26]. Previous studies have confirmed that the presence of toxic materials can be minimized by extracting them with hot water brewing. Coffee waste residue is practically pure lignocellulose [27]. Consequently, scientists no longer consider these residues as a waste, but rather as a raw material for other processes [28]. 

Considering the literature and waste brewed coffee powder (WBCP) availability in massive amounts, WBCP is a qualified raw material as a carbon material source for environmentally friendly treatments like pyrolysis [5,10]. The presented work demonstrated the applicability of a ‘best from waste’ approach. It can close the loop of brewed coffee waste by leading to a new porous carbon-based product. The porous surface of carbonized coffee waste is a unique property suitable for its high surface activity [29]. Thus, in this research, carbon from waste coffee was targeted and tested as a relative humidity sensor for industrial and household applications [30,31,32]. It has been found that the porous films exhibit higher humidity sensitivity than the nonporous counterparts [33]. The presence of inter/intragranular porosity and pore size distribution are influential factors for humidity sensors [34,35]. In this work the humidity sensing characteristics of pyrolyzed coffee waste-based sensors were investigated at room temperature in a relative humidity (RH) range from 0 up to 98%. 

## 2. Materials and Methods

### 2.1. Preparation and Characterization of Carbon from Waste Brewed Coffee Powder by Pyrolysis

A WBCP (Caffe Vergnano, Italy) was utilized as starting material for carbonization (Figure 1). The leftover chemicals in waste coffee powder after brewing may influence the natural structure of coffee during the carbonization step [36]. In order to remove the chemicals left over in waste coffee powder after brewing, the sampled WBCP was washed many times with water by centrifugation and then filtered. The WBCP was then dried in an oven at 90 °C for 10 h to evaporate completely the solvent. The pyrolysis of the material was then performed at 700 °C for 1 h in nitrogen atmosphere (120 mL/min) with 30 min dwells respectively at 250 °C and 400 °C. The heating ramp rate of the tubular furnace was set at 5 °C/min [37]. After the pyrolysis step, the material was manually ground to improve the homogeneity in the whole material.

Field Emission-Scanning Electron Microscope (FE-SEM, Zeiss Supra-40, Oberkochen, Germany) observations were carried out on the CGB powders with an IN-LENS detector equipped with a dispersive X-ray detector (Oxford Energy) while for low resolution and EDS we use an ETD detector (EVERHART-THORNLEY) and HV 12 to 25 Kv.

Particle size distribution was evaluated through a laser granulometer (Mastersizer 3000, Malvern, Worcestershire, UK). The measurements were carried out after dispersion in ethanol (500 ppm in volume) and sonication for 5 min.

Thermogravimetric-Differential Thermal Analysis TG-DTA (STA 409, Netzsch, Selb, Germany) was performed on CGB powder in the temperature range 25–700 °C with a heating ramp of 10 °C/min under static air. TGA was also performed on the WBCP under a 25 standard cubic centimeters per minute (SCCM) flow of argon in the same temperature range. 

A Renishaw micro-Raman (Renishaw, New Mills, UK) equipped with a green laser (514 nm) was utilized to study the disorder and graphitization grade of the CGB under test. Raman characterization were performed at room temperature. The samples, in powder form, were exposed, to the micro laser beam (2 µm in diameter) for 10 s without any pre-treatment.

X-Ray Diffraction (XRD) investigation was also performed on CGB powders and spectra were recorded on a Pan’Analytical X’Pert Pro instrument (Pan’Analytical, Almelo, The Netherlands) working with Cu K_ radiation (0.154056 nm) in the range 5–70° of 2θ, with a step size of 0.05° in 2θ. XRD analysis was carried out with the aim of evidencing any possible crystallization peak on CGB powder. Diffraction patterns were indexed by means of the Powder Data File database (P.D.F. 2000, International Centre of Diffraction Data, Newtown Square, PA, USA).

The chemical and electronical states of the CGB surface were studied by X-ray photoelectron spectroscopy (XPS) analysis, realized with a PHI 5000 Versaprobe (Physical Electronics, Chanhassen, MN, USA) scanning X-ray photoelectron spectrometer (monochromatic Al K-α source of X-ray with 1486.6 eV energy, 15 kV voltage, and 1 mA anode current).

### 2.2. Coffee Sensor Preparation

Resistive-type humidity sensors generally contain noble precious metal electrodes either deposited on a glass or ceramic substrate by thick or thin film deposition technique [38,39,40]. In this work, a commercial platinum ink (5545-LS, ESL, King of Prussia, PA, USA) was screen printed onto α-alumina substrates (Coors Tek, Golden, CO, USA, ADS-96, 96% alumina, 0.85 cm × 1.7 cm) obtaining interdigitated electrodes. These were subsequently fired at 980 °C (2 °C/min both heating and cooling ramps) with a dwell of 18 min, resulting in a high-degree of adhesion and optimizing their electrical conductivity. The interdigitated electrodes have a thickness of 6 µm and are spaced one from the other of 450 µm. Moreover, the edge of an electrode belonging to one comb and the vertical connection of the second comb are separated by 400 µm. An ink of CGB was prepared using ethylene glycol monobutyral ether, (Emflow, Emca Remex, Montgomeryville, PA, USA), as organic vehicle so that the proper rheological characteristics were provided to the ink after polyvinyl butyral (PVB) addition, which acts as temporary binder. In this study, the as-prepared CGB ink was used to screen print onto the alumina substrates equipped with Pt interdigitated electrodes by means of a 270-mesh steel screen, as shown in Figure 2. In order to evaluate the reproducibility of the as-fabricated CGB sensor, two sensors were prepared and tested. Screen printed sensors were heated at 300 °C in air for 1 h resulting in sensitive CGB thick film of 20.3 ± 1.6 μm.

### 2.3. Humidity Analysis Setup 

Relative humidity is defined as the ratio of the amount of moisture content of air to the maximum (saturated) moisture level that the air can hold at a same given temperature and pressure of the gas [33]. RH is a temperature dependent magnitude, and hence it is a relative measurement. The RH value (in %) is evaluated by the Equation (1):(1)$$RH\;\left(\%\right)=\frac{P_{v}}{P_{s}}\;\times \;100$$
where, P_v_ is the actual partial pressure of moisture content in air and P_s_ is the saturated pressure of moist air at the given temperature [33,41]. 

CGB sensors were placed in a laboratory apparatus (Figure 3) built with a hermetic chamber where relative humidity amounts can be adjusted between 0 and 98% by 12 steps of 3 min both in adsorption and in desorption to investigate the presence of a possible hysteresis between adsorption and desorption measurements. Test towards humidity were led under a constant flow of 1000 SCCM. In this system, compressed air flow (1) is split in two flows: one is desiccated over a chromatography alumina bed (2), the other goes through two water bubblers (3), generating a dry and a humid flow respectively. The two fluxes are recombined using two precision microvalves (4) in a unique flow with a mixer (5). A commercial probe for humidity and temperature measurements (9), Delta Ohm DO9406 (Caselle di Selvazzano (PD), Italy) was used as reference for temperature and RH values inside the chamber. During tests in a dynamic flow under variable amounts of humidity (RH) and target gases, the CGB sensors impedances were measured by means of LCR meter (Hioki 3533-01, Nagano, Japan), working with an AC voltage of maximum amplitude 1 V and frequency 1 kHz. 

Subsequently, the as-fabricated sensors’ selectivity was evaluated by exposing an already tested for relative humidity CGB sensor to ozone 200 ppb, nitrogen dioxide 200 ppb, ammonia 50 ppm and carbon dioxide 500 ppm (research quality gases, SIAD, Bergamo, Italy). All measurements were carried out at 27 ± 0.5 °C and under a constant flow of 1000 SCCM, and sensors were put in a three-neck glass flask chamber of 0.1 L volume. Dilutions of the target gas were performed with air by means of flow meters (Teledyne Hastings Instruments HFM 300 controller and flow meters HFC 302, Teledyne Hastings, Hampton, VA, USA). Ozone was generated by a UV lamp of Hg (SOG-01, UVP-LLC, Cambridge, UK) from a constant air flow of compressed air. Ozone concentration was determined by the length of lamp exposed, in accordance with the manufacturer’s calibration curves. 

The response times (that is the time that a sensor needs to reach the 90% of the total impedance change for the adsorption of gas) and the recovery times (the time taken for a sensor necessary to achieve the 90% of the total impedance changes in the case of gas desorption) were also calculated in the present work. 

The CGB sensor response (SR%) is calculated as follows (Equation (2)):(2)$$\mathrm{SR}\left(\%\right)=100\times \frac{\left|Z_{0}-Z_{g}\right|}{Z_{0}}$$
where, Z_0_ is the initial film impedance under dry air and Z_g_ is the impedance under humidity/gas exposure. 

## 3. Results

### 3.1. FESEM-EDS Characterization

According to the FE-SEM investigation in Figure 4, CGB powder possesses a high degree of porosity with cavities having an average diameter of around 7 µm. 

The agglomerates’ diameters are in the range between 10 and 40 µm, with small nanometric scale pores formed during the pyrolysis of WBCP. The overall structure seems very interesting and suitable for the adsorption of gas inside and outside of the channels. From EDX analysis carried out in three different regions of 25 µm^2^, the CGB chemical composition (semi-quantitative analysis) reveals a 95–96.8 wt.% of carbon, with oxygen concentration between 2.4 and 3.1 wt.% and traces of other elements like Mg, P and K still present after the pyrolysis. Laser granulometry of CGB powders evidenced that this biochar exhibits a bimodal distribution with two maxima at 23 and 192 µm (in Appendix A). The size distribution is in agreement with the FESEM observations. 

### 3.2. TGA Analysis

To study the carbonization process and to set the pyrolysis parameters, thermogravimetric analysis (TGA) under an argon flow (100 SCCM) was performed. The heating ramp of the furnace was maintained at 5 °C/min until 700 °C. The pyrolysis conditions were maintained for 1 h at 700 °C. The TG-DTG plots are shown in Figure 5.

The total weight loss observed was 82%, in accordance with the yield of WBCP pyrolysis that was 18 wt%. The first derivative confirms three exothermic peaks at 66 °C, 305 °C and 388 °C. Maximum weight loss rate was observed at 305 °C.

The pyrolysis process took place in three decomposition stages. The first one occurred from 25 to 200 °C, corresponding to a small mass loss of 9% due to dehydration and a slight release of volatile organic compounds. The second one, ranging between 200 to 500 °C, was associated to the major mass loss of 71%, that is attributed to devolatilization, including release from biomass degradation due to the decomposition of its main constituents (cellulose, hemicellulose and lignin [42]). The last step took place from 500 to 700 °C, due to the decomposition of the remaining biomass and the formation of the char. The DTA-TG curves of CGB powder are shown in Figure 6.

From the TG-DTA investigation under static air (Figure 6), at 700 °C, a 25% of mass loss was measured (in black), and from the DTA curve (in blue) a broad endothermic peak centered at 102 °C is due to the evaporation of free water, while a shoulder around 250 °C could be a consequence of the evaporation of molecular bound water. An even broader exothermic signal around 350 °C can be attributed to the combustion of the residual cellulose and hemicellulose by means of the formation and emission of volatile compounds [35].

### 3.3. Raman Analysis

In the Raman investigation, different regions of the CGB sample were studied and the results are exhibited in Figure 7 and Table 1.

The Raman analysis results is consistent with a sp^2^ carbon-based disordered structure. The two main regions of interest are 1000–1800 cm^−1^ (where the D and G peaks are found) and 2400–3200 cm^−1^ where the overtones of the first one is present (2D, D+G and 2G peaks). The first region cannot be properly fitted by using only two peaks, but rather four are needed, one for each Raman mode (see Table 1). This might suggest the coexistence of regions with different amount of disorder but in order to support such idea further investigation should be carried out, which are well beyond the scope of this paper. As per the overtone region it can be observed that although the signal is quite broad specific features are still present supporting the presence of more ordered regions with a predominance of disordered regions. In this region one peak for each mode was used in order to have information on the relative intensity of the contributions. The Id/Ig ratio value is 2.94 and all peak shapes are Gaussian as expected in a disordered system.

### 3.4. XRD

The amorphous properties of CGB are revealed by the XRD spectrum in Figure 8. The X-ray diffraction profile of the CGB powder exhibits a high background intensity indicating that the as-prepared powder contains highly disordered carbon with the characteristic broad humps at 2θ = 23.5° and 2θ = 43°, values typical of amorphous carbonaceous materials.

### 3.5. XPS Analysis

The electronic and chemical composition of the surface was studied by an XPS investigation. From the survey, the amount of O is higher compared to the EDX analysis, probably because CGB exhibits an enrichment of oxygen in the surface compared to the bulk. 

This analysis shows also the presence of the elements such as nitrogen (2.9 wt.%), phosphorous (0.5 wt.%) and calcium (0.9 wt.%) still present after the pyrolysis of WBCP. Moreover, from the high-resolution spectra of C1s peaks, it is possible to notice that disordered sp^2^ carbon is predominant with respect to the sp3 one (respectively equal to 67.8% at 284.8 eV and 16.2% at 286.2 eV of the total carbon). The ratio between sp^2^ and sp^3^ carbon is equal to 4.1. Peaks related to C-O at 288.3 eV, C=O at 290.5 eV and COOR at 293.1 eV are minor, accounting respectively to 9.3%, 2.8% and 3.9% of the total carbon. Considering the deconvolution of the oxygen peak in due signals, the peak at 532.8 eV due to C-O is higher (61.6% of total O) compared to the signal at 531.2 eV generated by C=O (38.4% compared to the overall O), in accordance with the high-resolution C peaks. XPS spectra are drawn in Figure 9. 

### 3.6. Sensor Response Characterization

The results of CGB sensor exposure towards different humidity levels are shown in Figure 10, where the impedance (Z) variation are depicted from 0% of relative humidity until 98% and SR% is calculated from Equation (2) in the same range of relative humidity. 

When CGB sensors are exposed to humidity at 27 °C, they behave like an n-type semiconductor. In fact, the SR% slope is positive, which means that the conductivity of the screen-printed films increases at higher humidity levels. More in detail, the sensor 1 response starts around 20% of relative humidity, while sensor 2 responds around 30% of RH, with SR% respectively equal to 51% and 61% under humid atmospheres (98–97% of RH). In the first case the impedance decreases from 25.2 MΩ in dry conditions until 12.3 MΩ, while in the second case the Z values drop from 23 MΩ in dry conditions until 8 MΩ in humid conditions. The high initial impedance value is probably due to the limited thickness of the sensing film (Figure 2).

The desorption process was also monitored and the maximum hysteresis of 15% in the sensor 1 response was measured under 74% of RH, confirming that the kinetics of desorption is comparable with respect to the adsorption one. Sensor response and recovery times were determined by exposing the CGB sensor after 1 year at different pulses of relative humidty, as illustrated in Figure 11.

From Figure 11, the CGB sensor exhibits a good repeatability between different pulses of humidity, with response and recovery times respectively equal to 15 and 20 s under 50% of relative humidity and 4.5 min and 1 min under 90% of relative humidity. These short values are probably due to the considerable porosity of the film characterized by long channels where gas molecules can easily enter being physiosorbed onto the biochar carbon-based surface. In addition, aging of the sensor was evaluated after 1 year of storage at room temperature and the sensor performances were comparable at higher humidity value (42 vs 39% of impedance variation) but the aged sensor exhibits a decrease in the sensor response under 50% of relative humidity (from 14 to 9% of SR%) probably as a consequence of the degradation of some organic groups present on the CGB surface.

Furthermore, cross-sensitivity tests were carried out on the CGB sensor in order to evaluate the selectivity of the film for humidity and the results are displayed in Figure 12.

Generally, negligible interferences were detected for carbon dioxide 500 ppm, ozone 200 ppb, nitrogen dioxide 200 ppb and ammonia 50 ppm, as depicted in Figure 12. 

## 4. Discussion

It is well known that the presence of defects in the carbon surface improves the sensor response towards humidity [43,44] and the CGB sensor exhibits a high ratio sp^2^/sp^3^, as confirmed by our Raman and XPS analyses. In fact, a value equal to 4.2 for this parameter was calculated from XPS investigation. These defects act also as adsorption sites for water molecules. Moreover, the presence after the pyrolysis of oxygen derivatives improve the hydrophilic properties of the carbon surface [36]. When the RH values are above of 20–30%, an electronic transfer is probably involved between water molecules and the disordered carbon structure of the biochar grains.

The CGB sensor morphology is probably characterized by connected channels, forming a wide number of paths through the tubes, called percolation paths. CGB has mesoporous structure with multi point calculated BET surface area = 5 m^2^/g and pore volume = 8.7 × 10^−3^ cc/g.

The water molecules act as reducing species and a charge transfer from H_2_O to CGB causes a shift of the valence bond of CGB away from the Fermi level, resulting in an electron accumulation layer that decreases the impedance values when increasing the H_2_O concentration [37].

The exponential trend of the sensor response with RH amounts could be a consequence of the fact that up to 60% of RH more water molecules are adsorbed on the surface at room temperature, forming clusters of multi-layer of water molecules that are hydrogen-bonded [38]. In addition, these water molecules are able to condense into pores with size in the range 1–250 nm. At this point, the ionic conduction is predominant with respect to the electronic one after clusters of water formation and the hydration of protons into H_3_O^+^ is a thermodynamically favoured process in liquid water. Protons are the dominant charge carriers of water into the mesopores of the CGB sensor, leading to a fast increase in the layer conductivity at high humidity amounts.

In addition, the hydrophilic character is characterized by high values of sp^2^/sp^3^ ratio [45,46] equal to 4.18, determined by XPS investigation. To the best of our knowledge, only a few studies have reported the use of biochar materials for humidity monitoring at room temperature [5,45]. 

In Ziegler et al.’s [45] research, SWP700 (pyrolyzed mixed softwood pellets) and OSR700 (oil seed rape) commercial biochar were used as humidity sensitive materials, and both materials exhibited a p-type behaviour at low humidity values (between 5–25 RH% for SWP700 and in the range 5–40 RH% for OSR sensor). Moreover, the adhesion of the drop-coated thick film was improved by means of polyvinylpyrrolidone (PVP) creating a n-p heterojunction. In Afify et al.’s [5] research, pyrolyzed bamboo screen-printed thick-films were used as novel humidity sensors and impedance variation start around 10% of relative humidity. The features of the developed sensors are compared to the literature data on disordered carbonaceous materials in Table 2.

This work could open an interesting way of producing humidity sensors that are low cost, fast, environmentally friendly and selective for humidity measurements at room temperature. As a future perspective, possible ageing phenomena will be investigated by controlling if the sensor response towards humidity will be affected by storage after a long period of time.

## 5. Conclusions

In this work, coffee ground biochar (CGB) was evaluated as a novel humidity sensing material. The sensors were realized by a screen-printing technique, and the humidity sensing features were studied at room temperature in a relative humidity range from 0 to almost 100%. The as-realized CGB biochar was obtained from pyrolysis at 700 °C of waste brewed coffee powder (WBCP). The sensor impedance variation under humidity starts from 20% of relative humidity. The CGB exhibits a n-type semiconductor behaviour between 20–100 RH% with an exponential trend with increasing humidity amount. The response and recovery times were very fast under 50% of humidity (15–20 s), and the cross-sensitivity to carbon dioxide, ammonia, nitrogen dioxide and ozone was negligible.

## Figures and Tables

**Figure 1 sensors-19-00801-f001:**
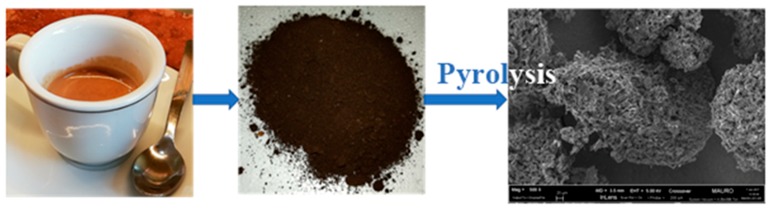
Conversion process of brewed coffee waste to carbon material.

**Figure 2 sensors-19-00801-f002:**
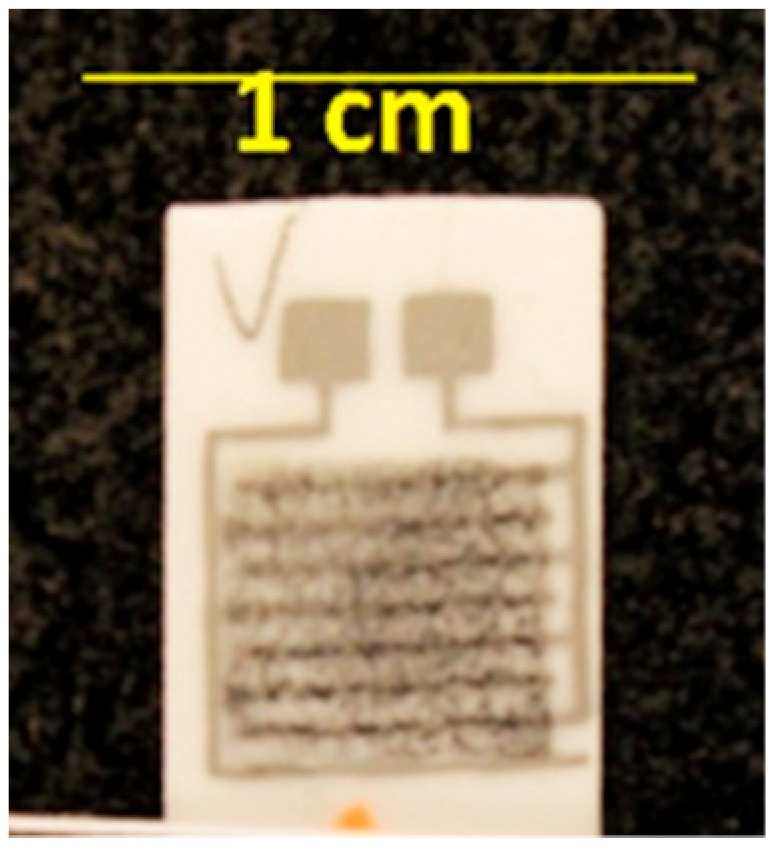
A picture of a CGB sensor screen printed sensor.

**Figure 3 sensors-19-00801-f003:**
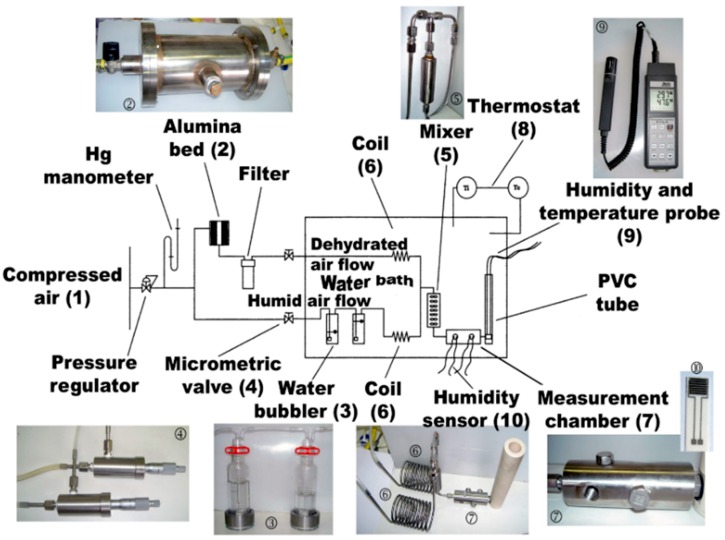
RH measurement system used in dynamic tests [33].

**Figure 4 sensors-19-00801-f004:**
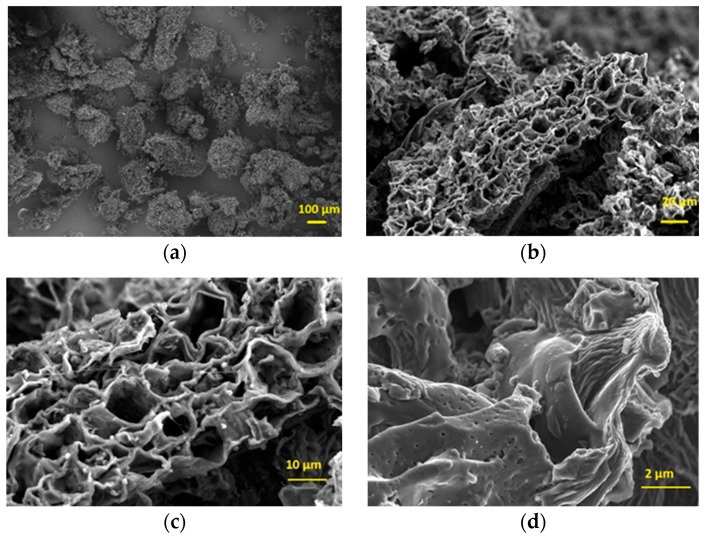
FE-SEM micrographs of the CGB powder at: (**a**) 150 ×, (**b**) 1 k ×, (**c**) 2.5 k × and (**d**) 15 k × magnification.

**Figure 5 sensors-19-00801-f005:**
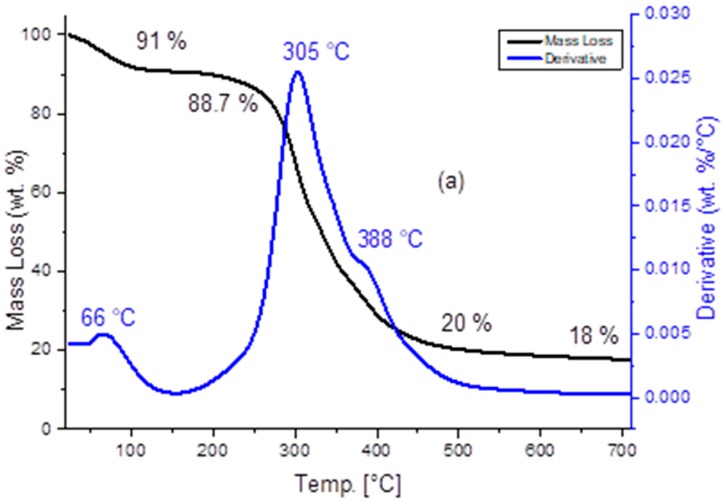
TGA of WBCP under argon.

**Figure 6 sensors-19-00801-f006:**
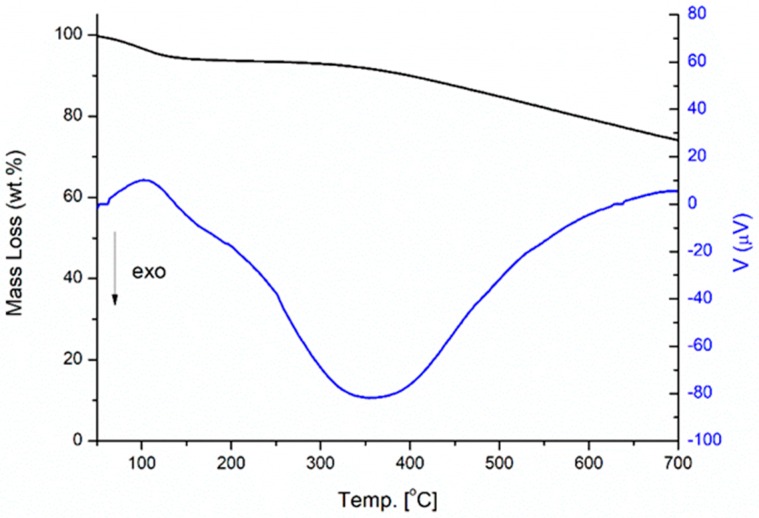
TG (black)-DTA (blue)- curve of CGB under static air.

**Figure 7 sensors-19-00801-f007:**
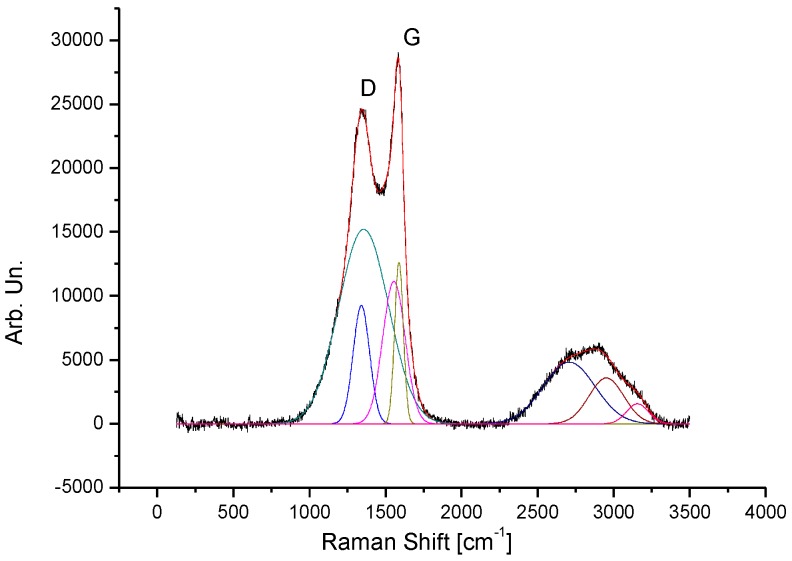
Raman spectrum of CGB.

**Figure 8 sensors-19-00801-f008:**
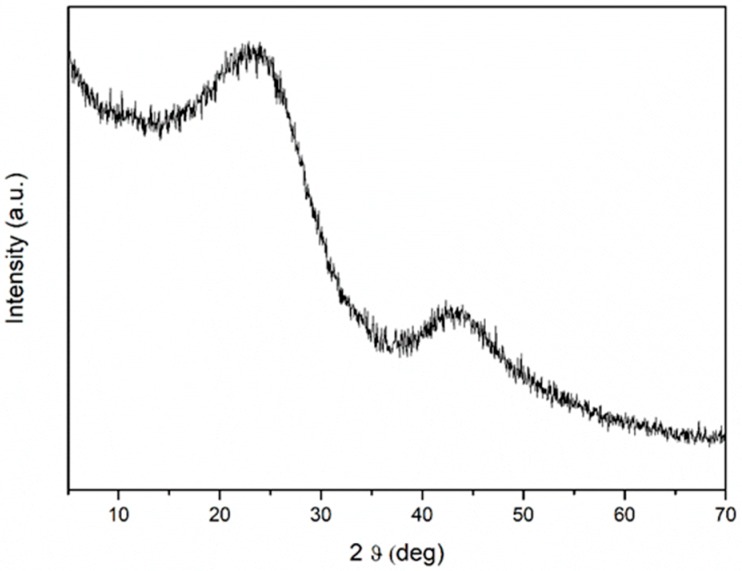
XRD pattern of CGB.

**Figure 9 sensors-19-00801-f009:**
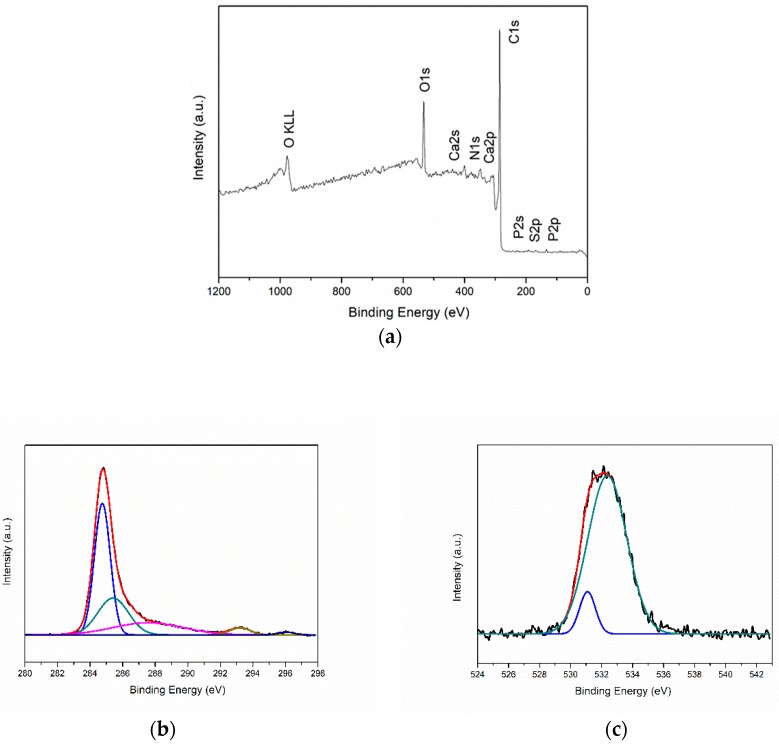
XPS spectra for CGB powder (**a**) survey, (**b**) high-resolution C and (**c**) high resolution O.

**Figure 10 sensors-19-00801-f010:**
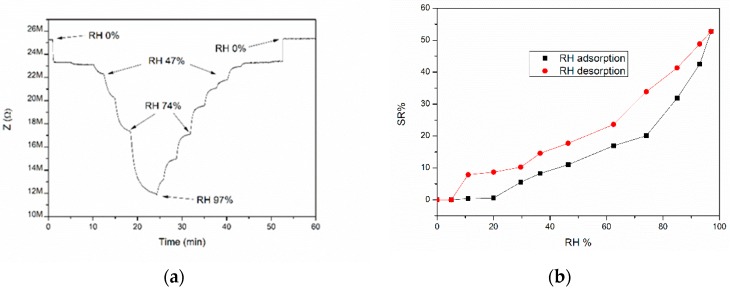
Impedance variation (**a**) and SR% values at different RH% (**b**) for CGB sensor during adsorption and desorption cycle.

**Figure 11 sensors-19-00801-f011:**
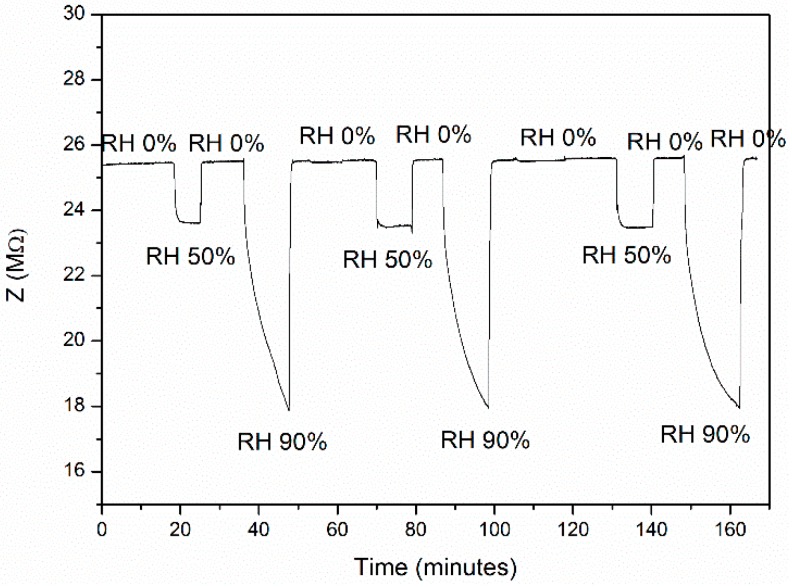
Impedance variation under 3 cycles of humidity pulses at 50% and 90% of relative humidity after 1 year of aging.

**Figure 12 sensors-19-00801-f012:**
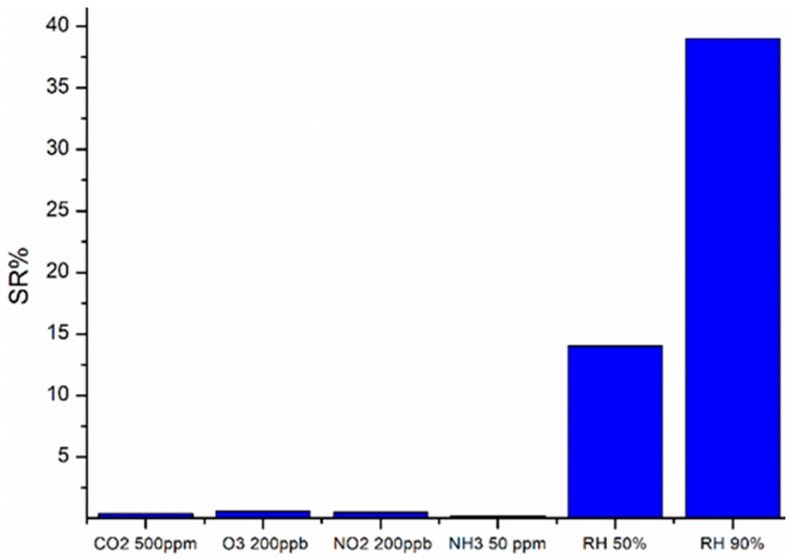
Cross sensitivity test of CGB sensor towards CO_2_ 500 ppm, O_3_ 200ppb, NO_2_ ppb and NH_3_ 50 ppm.

**Table 1 sensors-19-00801-t001:** Decomposition of peaks in Raman spectrum of CGB.

Label	Area [arb. un.]	Center [cm^−1^]	Width [cm^−1^]
D1	1.2	1341	54
D2	6.6	1365	166
G1	1.8	1559	68
G2	0.8	1589	28
2D	2.1	2708	168
D+G	0.8	2932	97
2G	0.5	3130	85

**Table 2 sensors-19-00801-t002:** Features of humidity sensors based on carbonaceous materials.

Sample	Resistance Change	Response Time	Recovery Time (s)	Ref.
C nanosheets produced by physical vapor deposition	Increase of 225% under 95 RH%	30 s when RH% increases from 11% to 40%	90 s when RH% decreases from 40% to 11%	[47]
Na-modified C films obtained by spray pyrolysis	Decrease of 97% under 60 RH%	n.d.	n.d.	[48]
Screen-printed commercial composite ink (ESL RS12113) made of epoxy resin and carbon powder	Increase of 4.8% under 80 RH%	n.d.	n.d.	[49]
Carbon quantum dots film made by electrochemical ablation of graphite	Resistivity decrease of 48% under 90 RH%	25 s when RH% increases from 7% to 43%	60 s when RH% decreases from 43% to 7%	[50]
Hydrogenated amorphous carbon (a-C:H) film	Decrease of 97.3% under 80 RH%	n.d.	n.d.	[51]
Pyrolyzed bamboo	Decrease of 91% under 95% RH	2 min	2 min	[5]
Pyrolyzed mixed softwood pellets	Decrease of 97.7% under 97.5% RH	1 min	1 min	[45]
Oil seed rape	Decrease of 94.5% under 99% RH	50 s	70 s	[45]
Coffee ground biochar	Decrease of 51% under 98%RH	4.5 min	1 min	This work

n.d.: not determined.

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
