# Peer review of "Waste Coffee Ground Biochar: A Material for Humidity Sensors"

_sensors, 2019, doi:10.3390/s19040801_

Round 1
Reviewer 1 Report
The title should be changed to humidity sensor. A list about different Biochars as humidity sensor ahould be given to provide more valuable results.
Author Response
The title should be changed to humidity sensor. A list about different Biochars as humidity sensor should be given to provide more valuable results.
We have implemented the suggested changes.
Reviewer 2 Report
The authors focused on the coffee waste as a source of biochar, that is coffee ground biochar (CGB). The authors synthesized CGB through the pyrolysis of coffee waste and fabricated humidity sensor by using the CGB as a sensing material. The synthesized CGB was characterized with FE-SEM, EDX, TGA, laser granulometry, TGA, Raman spectroscopy, XRD, and XPS. The sensing properties of the CGB-based humidity sensor was investigated by exposing it to different relative humidity ranging from 0% to 98%. The selectivity was also evaluated by exposing the sensor to other gas species.
Although many studies have reported the utilization of carbon materials for humidity sensors, the use of coffee waste for humidity sensors is new and interesting. The experiments are well organized and conducted. Thus, I recommend the paper for publication in Sensors. However, the following issues must be addressed before publication:
[Title] As this paper focuses on the application of CBG to humidity sensors, the title should contain “humidity” instead of “gas”.
[1. Introduction] Even though the authors mainly describe the issues related to coffee waste in the introduction, previous studies for humidity sensors that utilize carbon-based materials (including refs. 19 and 39) are also needed to be described in the introduction to clarify the novelty of the paper.
[3. Results] Several characterization results (Figs. 4-10) are not necessary for discussion; for example, the result of laser granulometry (Fig. 5) does not seem to provide any additional information. The authors should choose some important figures for the main script and put other figures in a supplementary material.
[3.4 Raman analysis] The Raman spectrum is fitted with 7 Gaussian peaks.
The authors should explain the scientific soundness of the fitting procedure. I think 7 peaks are excessive. The spectrum should be fitted with fewer peaks that can be assigned to specific vibration modes.
The values in the main script and Table 1 are not consistent: peaks at 1356, 1555, 1588 cm-1 [l. 225] are not listed in Table 1.
[3.6. XPS analysis] Higher resolution images should be provided for Fig. 10.
[l. 284] The authors need to add references which report that defects in carbon materials improve the sensor response to humidity.
[l. 303] The authors should explain why a high sp2/sp3 ratio leads to the hydrophilic character. The authors also add references for this phenomenon.
[l. 305] The performance of the CGB-based humidity sensors should be compared with the reported ones (i.e. SWP700 and OSR700 commercial biochar and pyrolyzed bamboo) in terms of some important factors such as sensing range, response/recovery time, and sensitivity. A table that summarizes these factors of the present and previous studies is preferable.
Author Response
The authors focused on the coffee waste as a source of biochar, that is coffee ground biochar (CGB). The authors synthesized CGB through the pyrolysis of coffee waste and fabricated humidity sensor by using the CGB as a sensing material. The synthesized CGB was characterized with FE-SEM, EDX, TGA, laser granulometry, TGA, Raman spectroscopy, XRD, and XPS. The sensing properties of the CGB-based humidity sensor was investigated by exposing it to different relative humidity ranging from 0% to 98%. The selectivity was also evaluated by exposing the sensor to other gas species.
Although many studies have reported the utilization of carbon materials for humidity sensors, the use of coffee waste for humidity sensors is new and interesting. The experiments are well organized and conducted. Thus, I recommend the paper for publication in Sensors. However, the following issues must be addressed before publication:
Thanks for your valuable and encouraging comments.
Point 1 [Title] As this paper focuses on the application of CBG to humidity sensors, the title should contain “humidity” instead of “gas”.
Response 1 Done.
Point 2 [1. Introduction] Even though the authors mainly describe the issues related to coffee waste in the introduction, previous studies for humidity sensors that utilize carbon-based materials (including refs. 19 and 39) are also needed to be described in the introduction to clarify the novelty of the paper.
Response 2 We completely agree on the comments and suggestion pointed out by reviewers. Following part on carbon roll in sensor mentioned in the introduction. Carbon might be the most widely-used material in sensor application. It has combined properties of the high surface area, electrical conductivity, biocompatibility, chemical and electrochemical stability. Many studies have shown that the electric properties of carbon nano-materials as sensors can efficiently sense the surface changes. The integration of CNTs into electrodes has been challenging due to cost and toxicity. On the contrary, biomass like coffee waste is a qualified carbon raw material for the synthesis of valuable carbon materials because it is available in high quality and huge amount, and it is an environmentally friendly renewable resource.
Point 3 [3. Results] Several characterization results (Figs. 4-10) are not necessary for discussion; for example, the result of laser granulometry (Fig. 5) does not seem to provide any additional information. The authors should choose some important figures for the main script and put other figures in a supplementary material.
Response 3 Thanks for your valuable suggestion.
Laser granulometry is describing the overall size distribution of biochar which might be interesting to define the activity of the sensor. FESEM images confirmed the laser granulometry observations. So, considering your comments we will keep the Granulometry graph in Supplementary material.
Point 4 [3.4 Raman analysis] The Raman spectrum is fitted with 7 Gaussian peaks.
The authors should explain the scientific soundness of the fitting procedure. I think 7 peaks are excessive. The spectrum should be fitted with fewer peaks that can be assigned to specific vibration modes.
Response 4 Considering your comments, we added the following statement to support the discussion we made in the article. Raman analysis is consistent with a sp2 carbon based disordered structure. The two main regions of interest are 1000-1800 cm-1 (where D and G peaks are found) and 2400-3200 cm-1 where the overtones of the first one is present (2D, D+G and 2G peaks). The first region cannot be properly fitted by using only two peaks, but four are needed, one for each Raman mode (see Table 1). This might suggest the coexistence of regions with different amount of disorder but in order to support such idea further investigation should be carried out, which are well beyond the scope to this paper. As per the overtone region it can be observed that although the signal is quite broad specific features are still present supporting the presence of more ordered regions with a predominance of disordered regions. In this region one peak for each mode was used in order to have an information on the relative intensity of the contributions. The Id/Ig ratio value is 2.94 and all peak shapes are Gaussian as expected in a disordered system.
Point 5 The values in the main script and Table 1 are not consistent: peaks at 1356, 1555, 1588 cm-1 [l. 225] are not listed in Table 1.
Response 5 This Part on Raman was recalculated and rearranged.
Point 6 [3.6. XPS analysis] Higher resolution images should be provided for Fig. 10.
Response 6 Done, after correction the figure 10 changes to Figure 9 in higher resolution.
Point 7 [l. 284] The authors need to add references which report that defects in carbon materials improve the sensor response to humidity.
Response 7 We added following two references support the statement mentioned in the article.
Llobet E., Gas sensors using carbon nanomaterials: A review. Sensor Actuators B Chem 2013, 179, 32-45.
Huang Q., Zeng D., Tian S., Xie C., Synthesis of defect graphene and its application for room temperature humidity sensing. Materials Letters 2012, 83, 76-79.
Point 8 [l. 303] The authors should explain why a high sp2/sp3 ratio leads to the hydrophilic character. The authors also add references for this phenomenon.
Response 8 The hydrophilic character is evidenced by a higher sp2/sp3 ratio in accordance with the literature as well as our co-authors previous experience on other biochar which they already demonstrated in the past. The published articles by them as well as one more article added as a reference for this statement.
Ziegler D., Palmero P., Giorcelli M., Tagliaferro A. Tulliani J.-M. Biochars as Innovative Humidity Sensing Materials. Chemosensors 2017, 5, 35.
Paul R. et al, Synthesis of DLC films with different sp2 /sp3 ratios and their hydrophobic behaviour, J. Phys. D: Appl. Phys. 2008, Volume 41, Number 5,55309.
Point 9 [l. 305] The performance of the CGB-based humidity sensors should be compared with the reported ones (i.e. SWP700 and OSR700 commercial biochar and pyrolyzed bamboo) in terms of some important factors such as sensing range, response/recovery time, and sensitivity. A table that summarizes these factors of the present and previous studies is preferable.
Response 9 To this aim, Table 2 was added to the text where disordered carbonaceous sensors were compared to the proposed sensor.
Reviewer 3 Report
This is a rather interesting article, since unconventional sensitive material has been used. However, the article requires some refinement. The authors must take into account the following comments.
1. In Introduction the authors postulate that "High surface area and porosity of carbonized coffee waste are unique properties suitable for the high surface activity". However there is no information about the specific surface area of the sensitive material obtained in this work. This information must be added to the text of the article.
2. The FTIR investigations are necessary for the characterization of surface active groups.
3. The figures with O1s and C1s XPS data (Figure 10 a,b) are of very low quality. It is necessary to redraw these figures.
4. In lines 265-267 the authors wrote: "More in detail, sensor 1 response starts around 20% of relative humidity, while sensor 2 around 30% of RH, with SR% respectively equal to 51% and 61% under humid atmospheres (98-97% of RH)." Thus, the question arises about the reproducibility of the properties of the sensors and the stability of their work for a long time. These parameters have to be discussed in the article.
Author Response
This is a rather interesting article, since unconventional sensitive material has been used. However, the article requires some refinement. The authors must take into account the following comments.
Point 1 In Introduction the authors postulate that "High surface area and porosity of carbonized coffee waste are unique properties suitable for the high surface activity". However, there is no information about the specific surface area of the sensitive material obtained in this work. This information must be added to the text of the article.
Response 1 The measured BET information was added in the statement (please check the line 312) mentioned in the article as follows, CGB has mesoporous structure with multi point calculated BET surface area = 5 m2/g and pore volume = 8.7 * 10-3 cc/g.
Point 2 The FTIR investigations are necessary for the characterization of surface-active groups.
Response 2 Due to the absorbing nature of carbon, measurements in transmittance mode FTIR are not informative. The same is true for ATR analysis as it requires a long path in the material. We gave a few tries but no reliable signal was obtained. For this reason, we privileged XPS in order to investigate the surface of the material.
Point 3 The figures with O1s and C1s XPS data (Figure 10 a, b) are of very low quality. It is necessary to redraw these figures.
Response 3 Done, and figure 10 is the new figure 9.
Point 4 In lines 265-267 the authors wrote: "More in detail, sensor 1 response starts around 20% of relative humidity, while sensor 2 around 30% of RH, with SR% respectively equal to 51% and 61% under humid atmospheres (98-97% of RH)." Thus, the question arises about the reproducibility of the properties of the sensors and the stability of their work for a long time. These parameters have to be discussed in the article.
Response 4 We have investigated the repeatability of the sensor 1 by exposing the sensor at different humidity levels (50 and 90%) for three pulses and the curves are almost superimposable. Nevertheless, the reproducibility should be improved since the 2 sensors tested exhibit different sensitivity towards humidity. Finally, the stability of their work was evaluated after 1 year of aging and the sensor response was comparable at higher humidity values but decrease from 14% to 9% under 50% of relative humidity probably as a consequence of the low stability of the organic groups present on the surface of CGB sensor, confirmed in the XPS analysis.
Reviewer 4 Report
In this paper authors presented carbon structure for humidity sensing application. The structure was obtained from coffee waste via pyrolysis at 700oC. Material was applied on the interdigitated transducer to obtain resistance/impedance gas sensor. Structure was extensively characterized using methods like: SEM, Raman, XRD and XPS. Also the base material - coffee waste and product (CGB) thermal behavior was tested using termogravimetrical methods. Authors tested obtained device (sensor) to RH changes in the air, and also checked its selectivity to other gases like O3, NH3 and CO2.
In my opinion manuscript is very interesting and authors idea seems to be novel. Of course carbon materials are used in gas sensing from many years but to my best knowledge presented structure obtained from coffee waste was not presented before in gas sensing application.
The introduction part is interesting but finally the sensing material is carbon microstructure, so in my opinion some information about carbon materials used as gas sensing have to be presented. Structures like graphite, graphite oxide, graphene, graphene oxide, carbon nanotubes and other carbon micro/nano structures should be mentioned in the introduction to show proper state of the art. To be honest also presented motivation - the management of waste from coffee, is doubtful because very small amount of coffee waste is necessary to obtain such a sensor what is more a huge amount of energy is necessary to carry out the pyrolysis process. Please imagine how huge should be the production of such devices to utilize 1% of world coffee waste and how many energy it will consume.
The experimental part is well written and clear, however I would like to ask why such huge gas flow (1700 SCCM) was chosen? I am also confused, what was the gas flow during humidity tests 1700 SCCM (page 4 line 142) or 1000 SCCM (page 5 line 156)? In my opinion it would be valuable to show how the sensor works in alternating dry / humid cycles - now it is impossible to check the response/recovery parameters of the sensor. The reposnse and recovery times should be checked from baseline signal (for example 0% RH) to signal in measured RH value. Why authors chose such small concentrations of NO2, Ozone and CO2 - only 200 ppb, for selectivity study? Is used frequency 1 kHz optimal for humidity sensing?
The material characterization presented in results is strong side of the paper, authors showed the morphology, chemical composition, particle size, crystalline properties (amorphous structure), and thermal stability of the produced sensing material. In this part I have few technical points:
- what electrons energy (accelerating voltages) and detectors were used in FE-SEM measurements.
- table 1: units are missing for center Raman Shift and peak width.
- figure 10b and 10c should be presented in better resolution/quality - they are not readable.
- in figure 11a please mark clearly the RH values for all cases not only of 0% and 97%.
- in figure 11 b maximum response value is about 50% (0.5) but in the fig 12 log(SR%) for RH obtained values like 10 or 40 (for 50%RH and 90% RH respectively). Please explain how is it possible if log(50) is equal about 1.7?
- did authors measured sensor response to alternate dry/humid/dry/humid.... cycles? As I mentioned before, it would be good to show such a result to characterize response and recovery times, repeatability etc..
Discussion part:
-Page 12 line 286 what did authors mean in the phrase "(4.2 in the latter analysis)"?
- please compare sensing parameters of presented material to other carbon materials more in detail (responses, concentration range, response/recovery times) please compare also the results to carbon materials different than biochar ones.
- how many devices did authors manufactured and tested? what is the repeatability of the proposed sensors?
Summarizing, the paper is interesting consist elements of novelty and fits to Sensors journal scope, however some major improvements of the manuscript have to be done before the publication this is why my recommendation is major revision of the manuscript.
Author Response
In this paper authors presented carbon structure for humidity sensing application. The structure was obtained from coffee waste via pyrolysis at 700 oC. Material was applied on the interdigitated transducer to obtain resistance/impedance gas sensor. Structure was extensively characterized using methods like: SEM, Raman, XRD and XPS. Also, the base material - coffee waste and product (CGB) thermal behaviour was tested using termogravimetrical methods. Authors tested obtained device (sensor) to RH changes in the air, and also checked its selectivity to other gases like O3, NH3 and CO2. In my opinion manuscript is very interesting and authors idea seems to be novel.
Of course, carbon materials are used in gas sensing from many years but to my best knowledge presented structure obtained from coffee waste was not presented before in gas sensing application.
Thanks for the comments
Point 1 The introduction part is interesting but finally the sensing material is carbon microstructure, so in my opinion some information about carbon materials used as gas sensing have to be presented. Structures like graphite, graphite oxide, graphene, graphene oxide, carbon nanotubes and other carbon micro/nano structures should be mentioned in the introduction to show proper state of the art.
Response 1 We completely agree on the comments and suggestion pointed out by reviewers. Following part on carbon roll in sensor mentioned in the introduction. Carbon might be the most widely-used material in sensor application. It has combined properties of the high surface area, electrical conductivity, biocompatibility, chemical and electrochemical stability Many studies have shown that the electric properties of carbon nano materials as a sensor electrode exhibit efficient sense characteristics. The integration of CNT into electrodes has been challenging due to cost and toxicity. In contrary, biomass like coffee waste is a qualified carbon raw material for the synthesis of valuable carbon materials because it is available in high quality and huge amount, and it is an environmentally friendly renewable resource.
Point 2 To be honest also presented motivation- the management of waste from coffee, is doubtful because very small amount of coffee waste is necessary to obtain such a sensor what is more a huge amount of energy is necessary to carry out the pyrolysis process. Please imagine how huge should be the production of such devices to utilize 1% of world coffee waste and how much energy it will consume.
Response 2 I completely agree with reviewer’s comments, the amount of quantity required for this kind of studies are limited. In this study, the effort was taken to demonstrate the sensitivity of the material. In our research group we are focusing our further research goal on use of brewed wastes coffee for a sensor in civil and construction applications. Finally, considering the production of CGB in an electrical furnace, the consumed electricity could be produced from renewable resources.
Point 3 The experimental part is well written and clear, however I would like to ask why such huge gas flow (1700 SCCM) was chosen? I am also confused, what was the gas flow during humidity tests 1700 SCCM (page 4 line 142) or 1000 SCCM (page 5 line 156)? In my opinion it would be valuable to show how the sensor works in alternating dry / humid cycles - now it is impossible to check the response/recovery parameters of the sensor. The response and recovery times should be checked from baseline signal (for example 0 % RH) to signal in measured RH value. Why authors chose such small concentrations of NO2, Ozone and CO2 - only 200 ppb, for selectivity study? Is used frequency 1 kHz optimal for humidity sensing?
Response 3 The first measurements were carried out in an old system under a fixed gas flow of 1700 SCCM, and then a new system was realized for cross sensitivity tests under different gases and gas flows. For this reason, we tested the sensors towards carbon dioxide, ozone, nitrogen dioxide and ammonia under 1000 SCCM. However, we retested the sensor at different humidity levels in the new system under 1000 SCCM and no effects on the gas flow was noticed in this test. Moreover, sensor response and recovery times were determined by exposing the CGB sensor under 50 and 90% of relative humidity for three times and the repeatability was acceptable as depicted in the new figure 11. We selected those gas concentrations because the threshold limits above which they can be hazardous for the human health were considered. Finally, we have not investigated the effect of the frequency on the sensor response, and 1kHz was selected to avoid any polarization phenomena than can occur at low frequencies when sensors are exposed to H2O.
Point 4 The material characterization presented in results is strong side of the paper, authors showed the morphology, chemical composition, particle size, crystalline properties (amorphous structure), and thermal stability of the produced sensing material. In this part I have few technical points:
- what electrons energy (accelerating voltages) and detectors were used in FE-SEM measurements.
Response 4 Usually for high resolution, we use IN-LENS detector and 3 to 5 kV, while for low resolution and EDS we use ETD detector (EVERHART-THORNLEY) and 12 to 25 kV. We added this information in line 93 and 94.
Point 5- table 1: units are missing for centre Raman Shift and peak width.
Response 5 Corrected
Point 6- figure 10b and 10c should be presented in better resolution/quality - they are not readable.
Response 6 Done in the new figure 9.
Point 7- in figure 11a please mark clearly the RH values for all cases not only of 0% and 97%.
Response 7 Done, in the new figure 10
Point 8- in figure 11 b maximum response value is about 50% (0.5) but in the fig 12 log (SR%) for RH obtained values like 10 or 40 (for 50%RH and 90% RH respectively). Please explain how is it possible if log (50) is equal about 1.7?
Response 8 The idea was to present the sensor responses SR% in logarithmic scale in order to better distinguish the SR% values for carbon dioxide, ozone, nitrogen dioxide and ammonia. Anyway, we have modified the graph in the figure 12 since in this way the higher sensitivity for humidity compared to other gas can be observed for CGB sensor.
Point 9- did authors measured sensor response to alternate dry/humid/dry/humid.... cycles? As I mentioned before, it would be good to show such a result to characterize response and recovery times, repeatability etc.
Response 9 We have investigated the repeatability of the sensor 1 by exposing the CGB sensor at different humidity levels (50 and 90%) for three pulses and the curves are almost superimposable. Nevertheless, the reproducibility should be improved since the 2 sensors tested exhibit different sensitivity towards humidity as described in the text.
Discussion part:
Point 10-Page 12 line 286 what did authors mean in the phrase "(4.2 in the latter analysis)"?
Response 10- 4.2 is the ratio between Csp2 and Csp3 in the CGB material calculated from the peak intensities after deconvolution for carbon in XPS investigation. We modified it properly in the text.
Point 11 Please compare sensing parameters of presented material to other carbon materials more in detail (responses, concentration range, response/recovery times) please compare also the results to carbon materials different than biochar ones.
Response 11 Table 2 was added to the text where disordered carbonaceous sensors were compared to the proposed sensor.
Point 12- how many devices did authors manufactured and tested? what is the repeatability of the proposed sensors?
Response 12 We have tested 2 CGB sensors towards humidity and reproducibility should be improved since the two sensors tested display different sensitivity towards humidity especially at low humidity values. In fact, in one case the impedance variation starts under 20% of humidity, in the other case under 30%. The differences are probably due to the different thickness of the two screen-printed films. Anyway, a difference of 20% in the response of screen-printed sensors is usually considered as normal. Nevertheless, the repeatability was evaluated and consistent for practical applications since the curves are superimposable when exposing the same sensor under 50% and 90% of humidity for three times, as depicted in the new figure 11.
Point 13 Summarizing, the paper is interesting consist elements of novelty and fits to Sensors journal scope, however some major improvements of the manuscript have to be done before the publication this is why my recommendation is major revision of the manuscript.
Response 13 Thanks for valuable comment and suggestions. It helped us to present our study in more informative way than before.
Round 2
Reviewer 2 Report
The authors have addressed all the issues I suggested.
The paper can be accepted after some minor corrections as listed below:
[l. 48] The reference is missing.
[Equations] Subscriptions should be used for equations; for example, Zg should be replaced by Zg.
[Fig. 8] There is a spelling error in the y-axis title.
Author Response
The authors have addressed all the issues I suggested.
The paper can be accepted after some minor corrections as listed below:
Point 1. [l. 48] The reference is missing.
Response 1. [l. 48] No.14 reference is added in the missing place.
Point 2. [Equations] Subscriptions should be used for equations; for example, Zg should be replaced by Zg.
Response 2. DONE, and reported corrections are mentioned on [line 177]
Point 3. [Fig. 8] There is a spelling error in the y-axis title.
Response 3. Corrected.
Reviewer 3 Report
The authors included information on the specific surface area (5 m2 / g). This value can not be attributed to the "high specific surface area." Therefore, it is necessary to rewrite theAbstract and Introduction, removing from there the argument that such materials are characterized by a high specific surface area (lines 18, 38, 72).
The quality of figures 9b and 9c is still unsatisfactory. These figures need to be rebuilt using any program that allows processing digital data (as is done for Raman spectroscopy data in Fig. 7).
Author Response
Point 1. The authors included information on the specific surface area (5 m2 / g). This value cannot be attributed to the "high specific surface area." Therefore, it is necessary to rewrite the Abstract and Introduction, removing from there the argument that such materials are characterized by a high specific surface area (lines 18, 38, 72).
Response 1. As per the reviewer’s comments and suggestions, the wording ‘Porosity and High surface area’ is replaced with ‘POROUS SURFACE’ in the indicated lines.
Point 2. The quality of figures 9b and 9c is still unsatisfactory. These figures need to be rebuilt using any program that allows processing digital data (as is done for Raman spectroscopy data in Fig. 7).
Response 2. Corrected.
Reviewer 4 Report
The authors addressed all my comments and improved the manuscript in satisfying way. From my point of voe this paper may be published in Sensors in current form.
Author Response
The authors addressed all my comments and improved the manuscript in satisfying way. From my point of view this paper may be published in Sensors in current form.
I thank you for your valuable suggestions which helped us to rectify our mistakes.